# Linear Tumor Regression of Rectal Cancer in Daily MRI during Preoperative Chemoradiotherapy: An Insight of Tumor Regression Velocity for Personalized Cancer Therapy

**DOI:** 10.3390/cancers14153749

**Published:** 2022-08-01

**Authors:** Soo-Yoon Sung, Sea-Won Lee, Ji Hyung Hong, Hye Jin Kang, So Jung Lee, Myungsoo Kim, Ji-Hoon Kim, Yoo-Kang Kwak

**Affiliations:** 1Department of Radiation Oncology, Eunpyeong St. Mary’s Hospital, Catholic University of Korea College of Medicine, Seoul 06591, Republic of Korea; ssy-uni@hanmail.net (S.-Y.S.); lords_seawon@hotmail.com (S.-W.L.); 2Division of Medical Oncology, Department of Internal Medicine, Eunpyeong St. Mary’s Hospital, Catholic University of Korea College of Medicine, Seoul 06591, Republic of Korea; jh_hong@catholic.ac.kr; 3Department of Radiation Oncology, Incheon St. Mary’s Hospital, Catholic University of Korea College of Medicine, Seoul 06591, Republic of Korea; jade2959@hanmail.net (H.J.K.); sunport@naver.com (S.J.L.); mskim0710@gmail.com (M.K.); 4Department of Surgery, Incheon St. Mary’s Hospital, The Catholic University of Korea, Seoul 06591, Republic of Korea; samryong@catholic.ac.kr

**Keywords:** rectal cancer, preoperative concurrent chemoradiotherapy, daily tumor regression, tumor volumetry

## Abstract

**Simple Summary:**

We primarily assessed the pattern of tumor regression throughout the full course of CCRT using daily registration MRI (25 sets of MRI per patient) and, for the first time, observed in vivo linear tumor regression in daily MRI. Secondly, we investigated if the velocity of tumor regression had an effect on DFS of locally advanced rectal cancer and found its potential as a biomarker for real-time treatment response and further treatment decision. We believe this study makes a significant contribution to the literature compared to previous reports, especially by first demonstrating the linear pattern of daily tumor regression during neoadjuvant CCRT in an in vivo setting. We also suggested the potential value of tumor regression velocity as a marker for further treatment decisions in a disease with diverse treatment options, such as locally advanced rectal cancer.

**Abstract:**

Objective: Neoadjuvant chemoradiotherapy (CCRT) is current standards of care for locally advanced rectal cancer. The precise and thorough investigation of a tumor during the full course of CCRT by means of daily MRI can provide an idea on real-time treatment sensitivity in addition to tumor biology. Tumor volumetry from daily MRI during CCRT may allow patient-driven treatment decisions. Material and Methods: Patients diagnosed with cT3-4 and/or cN+ rectal adenocarcinoma undergoing preoperative CCRT with capecitabine on the pelvis up to 50 Gy in 25 daily fractions from November 2018 to June 2019 were consecutively included. Rectal tumor volume was uniformly measured by a single physician (YKK) in daily 0.35T MRI obtained with MR-guided linear accelerator. Primary endpoint was to assess the pattern of tumor volume regression throughout the full course of CCRT using daily registration MRI. Secondary endpoint was to assess the effect of tumor regression velocity on disease-free survival (DFS). Tumor regression velocity (cc) per fraction of each patient was calculated using the simple regression analysis of tumor volumes from fraction 1 to fraction 25. Results: Twenty patients were included. Daily tumor volumetry demonstrated linear tumor regression during CCRT. The tumor regression velocity of all 20 patients was 2.40 cc per fraction (R^2^ = 0.93; *p* < 0.001). The median tumor regression velocity was 1.52 cc per fraction. Patients with tumor regression velocity ≥ 1.52 cc per fraction were grouped as rapid regressors (N = 9), and those with tumor regression velocity < 1.52 cc per fraction were grouped as slow regressors (N = 11). Rapid regressors had greater tumor regression velocity (4.58 cc per fraction) compared to that of slow regressors (0.78 cc per fraction) with statistical significance (*p* < 0.001). The mean DFS of rapid regressors was 36.8 months, numerically longer than the 31.9 months of slow regressors (*p* = 0.400) without statistical significance. Rapid regressors had numerically superior DFS rate compared to slow regressors without statistical significance. The 2-year DFS was 88.9% for rapid regressors and 72.7% for slow regressors, respectively (*p* = 0.400). Conclusion: This study is the first observation of linear tumor regression in daily MRI during the preoperative CCRT of locally advanced rectal cancer. Daily tumor regression velocity discriminated DFS, although without statistical significance. This study with a phenomenal approach is hypothesis-generating. Nevertheless, the potential of CCRT from therapeutics to a newer level, the “theranostics”, has been inceptively suggested. Further validation studies for the value of daily tumor volumetry on treatment decisions are warranted.

## 1. Introduction

Colorectal cancer is globally the third most common malignancy with nearly two million new cases per year [1]. It is the second most common cause of cancer death, resulting in nearly a million deaths worldwide [1]. Although the survival rate in early stage is high, approximately 80% [2], loco-regionally advanced stage requires multimodal therapy both pre- and post-surgery in order to improve outcome. The advancement in treatment strategies has been remarkable in the field of rectal cancer, establishing neoadjuvant concurrent chemoradiotherapy (CCRT) as the current standards of care for the locally advanced disease [3,4,5,6,7]. The advances are still in progress and have developed in both ways: treatment intensification, such as total neoadjuvant therapy, and treatment de-intensification, such as the watch-and-wait approach [8,9].

Total neoadjuvant therapy (TNT) administers neoadjuvant chemotherapy in addition to the standard neoadjuvant concurrent chemoradiotherapy for the further downstaging of tumor, earlier control of occult micrometastatic disease, and evaluation of chemosensitivity to initial regimen [10]. In a meta-analysis of landmark studies, pathologic complete response (odds ratio [OR] 2.44; *p* < 0.001) and disease-free survival (OR 2.07; *p* = 0.009) both significantly favored TNT over current standard care [8]. Although a national registry study suggested microsatellite instability (MSI) as a marker for patients who are more likely to benefit from TNT [11], contradicting results are also reported [12].

On the other side of the spectrum of rectal cancer treatment is the watch-and-wait approach. The omission of surgery after the achievement of clinical complete remission (cCR) allows better quality of life by lowering morbidity. Even before the introduction of TNT, the watch-and-wait approach has been reported to have an encouraging disease-free survival up to 84% at 10 years of follow-up [13]. After TNT, a retrospective analysis reported a sustained cCR rate of 22% at 1 year [14]. However, a careful selection of candidates for omission of curative surgery is imperative. The current evidence for patient selection of the watch-and-wait approach is low level because most studies on this topic include patients who refuse surgery, thus are based on heterogeneous, retrospective data.

In short, the standards for patient selection are unknown in both treatment intensification as well as de-intensification. These criteria for patient selection require being provided as early as possible in the course of cancer management so that they can guide the subsequent treatment decision. Because true endpoints, such as disease-free survival (DFS) and overall survival (OS), require long-term follow-up, a surrogate endpoint, such as pathologic response after neoadjuvant therapy, has been actively explored and demonstrated good correlation with true endpoints [15,16,17]. However, pathologic response can be assessed only after surgery and then it would be too late to provide information on whether to omit surgery or not.

Given this context, another marker under active investigation is the tumor response to CCRT in imaging before surgery. The majority of the previous studies exploring tumor volume change after preoperative CCRT use magnetic resonance imaging (MRI) acquired after CCRT just before surgery [18,19]. They report that the rate of tumor volume reduction obtained from MRIs at pre- and post-CCRT correlated with pathologic complete response. However, instead of relying on only two sets of MRI (pre- and post-CCRT), more information on tumor biology and treatment sensitivity can be obtained if greater number of observations are available [20]. For this reason, a Dutch group applied weekly MRI during CCRT to assess tumor volume change, in addition to two post-CCRT MRI [21]. With seven sets of MRI per patient, they attempted to extract a pattern of tumor response during CCRT and concluded that most of tumor regression occurred during the early half of CCRT course. Although daily observations are available with daily registration computed tomography (CT), the soft tissue contrast of cone beam CT is far inferior compared to that of MRI [22].

Against this background, we primarily aimed to assess the pattern of tumor regression throughout the full course of CCRT using daily registration MRI (25 sets of MRI per patient). Secondly, we investigated if the velocity of tumor regression had an effect on DFS of locally advanced rectal cancer. We anticipated that precise and thorough analysis of a tumor throughout the full course of CCRT with daily MRI may provide an idea on real-time treatment sensitivity and tumor biology, which may potentially guide patient-driven treatment decision.

## 2. Material and Methods

### 2.1. Patients

Patients diagnosed with locally advanced rectal cancer who underwent preoperative concurrent chemoradiotherapy (CCRT) at a tertiary medical institution from November 2018 to June 2019 were consecutively included. The inclusion criteria were as follows: (1) pathologically confirmed adenocarcinoma of the rectum by full colonoscopy; (2) clinical T3-4 tumors and/or positive lymph nodes (LN) according to the 8th edition of the American Joint Committee on Cancer TNM staging system; (3) completion of preoperative CCRT; (4) acquisition of daily registration MRI image before each session of radiotherapy; and (5) daily measurement of rectal tumor volume on registration MRI image. Patients with histology other than adenocarcinoma, distant metastasis at initial diagnosis, previous or concomitant malignancy of other primary origin, history of familial adenomatous polyposis or hereditary nonpolyposis colorectal cancer, and metallic foreign bodies or devices within the body contraindicated for MRI were excluded. All procedures of this study was approved by the institutional review board (IRB No. OC20RISE0036).

### 2.2. Diagnosis

Initial studies for diagnosis included physical examination; digital rectal examination (DRE); complete blood count; blood chemistry; carcinoembryonic antigen (CEA); computed tomography (CT) of chest, abdomen, and pelvis; full colonoscopy with rectal biopsy; rectal MRI; and position emission tomography (PET)-CT of torso. Diagnosis and treatment decisions were made in compliance with current cancer guidelines via the institutional multidisciplinary team, membered specifically for colorectal cancer by surgical, medical, and radiation oncologists, in addition to radiologic, nuclear medicine, and pathologic diagnosticians [23,24].

### 2.3. Treatment

Preoperative CCRT was given in a long course of 5 weeks in 5 days-a-week schedule with weekend breaks. For radiotherapy, patients were set up in supine position and were immobilized using Vac-Lok^TM^ (CIVCO Medical Solutions, Coralville, IA, USA). Simulation image was obtained using both the computed tomography (CT) for calculation of monitor unit (MU) for radiation delivery, as well as the MRI for improved soft tissue contrast during registration. CT simulation was performed using LightSpeed RT 16 (GE, Waukesha, WI, USA) with contrast enhancement (Optiray 320 Ultraject inj, 100 mL) except for patients with contraindications to contrast dyes. Simulation MRI image was acquired with ViewRay Linac MRIdian system (ViewRay, Cleveland, OH, USA) in sagittal, transverse, and coronal planes, and 4D images were also obtained. Target delineation was performed by board-certified radiation oncologists. Gross tumor volume (GTV) was defined as primary rectal mass and gross LNs. Clinical target volume (CTV) included mesorectum and pelvic lymphatics with pertinent extension according to involved LNs or adjacent organs, such as levator ani muscle, anal canal, or genitourinary tract [25]. Internal target volume (ITV) of rectal mass was created by adding 5−7 mm margin and encompassing involuntary rectal movement detected using 4D image of simulation MRI. PTV was created with a uniform margin of 3 mm to CTV for set-up error in addition to ITV. The GTV and PTV were used as the target for tracking and its safety margin, respectively. During radiation delivery, the beam automatically turned off if the target was out of the safety margin by a threshold of 3%.

Radiation was delivered up to a total of 50 Gy in 25 daily fractions to the whole pelvis, with boost to primary tumor at 46 Gy. Boost was also considered for clinically suspicious involvement of LNs outside mesorectum. Radiotherapy planning was performed by a board-certified medical physicist using MRIdian treatment planning system (ViewRay Inc., Mountain View, CA, USA).

MRI image was daily acquired during registration before delivery of radiotherapy for all patients using a 0.35T horizontal field MRI of ViewRay MRIdian Linac (ViewRay Inc., Oakwood, GA, USA). A whole-body radiofrequency (RF) transmit coil and surface receive coils located anterior and posterior to the patient were used. For torso, the receive coils were comprised of radiolucent phased arrays with 2 × 6 channels. A true fast imaging with steady stage precession (TRUFI) sequence was used for the pulse sequence to obtain volumetric and 4D MRI image. The field of view (FOV) with an in-plane resolution of 1.5 mm × 1.5 mm and slice thickness of 3 mm was used for volumetric imaging. For 4D MRI, the in-plane resolution of 3.5 mm × 3.5 mm with slice thickness of 5 mm was used. Registration was performed rigidly with axial, sagittal, and coronal image reconstructed from daily scanned image by shifting couch to accurately align the patient to the position from simulation image of the radiotherapy plan. Radiation was delivered when the patient position was within an acceptable range of <3 mm.

Concurrent chemotherapy regimen consisted of twice daily capecitabine with a dosage of 825 mg/m^2^ on days of radiation delivery. All patients were planned for radical resection and the type of surgery was left to the surgeons’ discretion. Adjuvant systemic therapy after surgery was administrated on the physicians’ discretion.

### 2.4. Follow-Up

Patients were scheduled for final preoperative MRI at 6 weeks after completion of CCRT. Clinical response was assessed using the response evaluation criteria in solid tumors version 1.1 [26]. Surgery was performed at approximately 8 weeks post-CCRT. After surgery, the patients were followed up every 3–6 months. At each follow-up, physical examination, DRE, blood tests, colonoscopy, CT, MRI, and/or PET-CT were performed at surgeon’s discretion.

### 2.5. Daily Tumor Volumetry

For consistency of tumor volume measurement, contour of rectal tumor on daily registration MRI image was manually delineated by single radiation oncologist (YKK) with over 10 years of experience. Tumor contour delineation was performed on each axial slice obtained from registration MRI image of each treatment session (Appendix A) using MIM Software version 6.9.8 (MIM Software Inc., Cleveland, OH, USA). Tumor volume (cc) within the delineated contour was volumetrically calculated using the same software.

### 2.6. Primary Endpoint

Primary endpoint was to assess the pattern of tumor volume regression throughout the full course of CCRT using daily registration MRI (25 sets of MRI per patient). Tumor regression velocity (cc) per fraction of each patient was defined as the slope of daily measured tumor volume, calculated using the simple regression analysis of tumor volumes from fraction 1 to fraction 25. The tumor regression velocity for all patients was calculated by the simple regression analysis of mean tumor volumes of all patients from fraction 1 to fraction 25.

### 2.7. Secondary Endpoint

Secondary endpoint was to assess the effect of tumor regression velocity on disease-free survival (DFS). The effect of tumor regression velocity on DFS was assessed using the Kaplan–Meier method with log-rank test.

### 2.8. Statistical Analyses

Mean ± standard deviation (SD), median, minimum, and maximum values of daily tumor volume (V_FxN_) measured at each fraction (FxN) were obtained for all patients. Tumor volume at baseline (V_Fx1_) was measured from the registration MRI image of the first fraction (Fx1) before delivery of radiation. Daily tumor volume (V_FxN_) was compared with baseline tumor volume (V_Fx1_) using the Wilcoxon’s signed rank test. Bonferroni correction was used for multiple comparison of 25 measurements, thus statistical significance was defined as *p* < 0.002 regarding tumor volume measurements. Regression rate (%) of tumor volume was calculated by dividing the difference of tumor volume from baseline at fraction 25 (V_Fx1_ − V_Fx25_) by baseline tumor volume (V_Fx1_) as follows: tumor regression rate (%) = (V_Fx1_ − V_Fx25_)/V_Fx1_ × 100.

Survival was calculated from the date of initiation of radiotherapy. Disease-free survival (DFS) was defined as the time interval to first detection of recurrence or death from any cause. Overall survival (OS) was defined as the interval to death from any cause. Survival rates were estimated using the Kaplan–Meier analysis with log-rank test. Continuous variables were compared with the Mann–Whitney test and categorical variables were compared with the Fisher’s exact test. Statistical significance was defined as *p* < 0.05. The statistical analyses were performed using the IBM SPSS (Statistical Package for Social Sciences) software for Windows, version 24.0 (IBM Corp., Armonk, NY, USA) and Microsoft Excel, version 2016 (Microsoft Corp., Redmond, WA, USA).

## 3. Results

### 3.1. Tumor, Patient, and Treatment Characteristics

Twenty patients satisfied the inclusion criteria. Table 1 shows tumor, patient, and treatment characteristics. When tumor regression velocity was obtained for each patient by calculating the slope of daily tumor volume of individual patient, the median value was 1.52 cc per fraction. The patients were divided into two arms, according to median tumor regression velocity (Table 1). Patients with tumor regression velocity ≥ 1.52 cc per fraction were grouped as rapid regressors (N = 9) and those with tumor regression velocity < 1.52 cc per fraction were grouped as slow regressors (N = 11). Rapid regressors had greater tumor regression velocity (4.58 cc per fraction) compared to that of slow regressors (0.78 cc per fraction) with statistical significance (*p* < 0.001, Table 1).

There was no difference in patient, tumor, and treatment characteristics between rapid and slow regressors (Table 1). The majority were male (90%) with a median age of 64. T3 tumors (65%) with nodal involvement (95%) were most common. There was no difference in clinical response between rapid and slow regressors. Clinical response after CCRT in MRI obtained before surgery for all 20 patients was as follows: complete response 5%, partial response 80%, and stable disease 15%. The preoperative MRI was obtained at median 5.7 weeks after completion of CCRT.

Among 20 patients who underwent preoperative chemoradiotherapy, only 14 patients had surgical resection of primary disease. Among 14 patients who underwent surgery, 9 had laparoscopic low anterior resection with ileostomy, 4 received intersphincteric resection, and 1 had conversion Hartmann’s operation. Six out of nine rapid regressors and eight out of eleven slow regressors underwent surgery. There was no difference in resection rate between rapid and slow regressors (Table 1).

There was no statistical difference in pathologic characteristics between rapid and slow regressors who underwent surgical resection (Table 1). The most predominant ypT stage for 14 patients who underwent resection was T3 (71.4%). The rate of ypT4 tumor was 7.1%, decreased from 35% of initial clinical T4 stage. The ypT1-2 tumors were observed in 21.4%. Nodal remission (ypN0) was achieved in 50%, and most of the node-positive disease was ypN1 (42.9%). Before CCRT, rapid regressors had more aggressive disease compared to slow regressors. Although statistically insignificant, rapid regressors had more T4 (rapid 44.4% vs. slow 27.3%) and more N2 (rapid 66.7% vs. slow 36.4%) disease. After CCRT, slow regressors had more aggressive features compared to rapid regressors, including more T3 (rapid 50% vs. slow 87.5%), more N1/2 (rapid 0% vs. slow 63.5%), and more vascular and perineural invasions, although without statistical significance.

### 3.2. Treatment Outcome

Median follow-up was 36 months (range: 11.7–40.5) for all 20 patients. Four patients had recurred (two local and two distant) and two patients had expired at the time of analysis. All recurrences were observed in the slow regressors except for one patient with a local recurrence at anorectal stump 8 months after surgery, who had a close surgical margin. Among two deaths, one was not related to cancer. For all 20 patients, median DFS was not reached and mean DFS was 34.6 months (standard error [SE]: 2.7, 95% confidence interval [CI]: 29.3–39.9). Median OS was not reached and mean OS was 38.6 months (SE: 1.4, 95% confidence interval [CI]: 35.8–41.5). For all 20 patients, 2-year DFS was 80% and 2-year OS was 95%, respectively. There was no difference in DFS (*p* = 0.730) between the resected patients (2-year 78.6%) and those who did not undergo resection (2-year 83.3%).

### 3.3. Primary Endpoint

Daily tumor volumetry demonstrated linear tumor regression during CCRT. When the mean tumor volumes (cc) of 20 patients who underwent neoadjuvant CCRT were plotted on daily basis, the pattern of tumor regression was linear, as shown in Figure 1. The tumor regression velocity of all 20 patients was 2.40 cc per fraction (R^2^ = 0.93; *p* < 0.001).

At baseline, mean tumor volume was 81.28 ± 48.77 cc. The significant regression of tumor volume (*p* < 0.0001) was observed as early as 4th fraction (mean 67.33 ± 40.55 cc) after three sessions of treatment. By the end of treatment, tumor volume was reduced to mean 23.04 ± 10.79 cc. The data of daily tumor volumetry for all patients are shown in Appendix A.

### 3.4. Secondary Endpoint

Tumor regression velocity (cc) per fraction, tumor regression rate (%), and DFS for all 20 patients grouped into rapid or slow regressors are shown in Table 2. Mean tumor regression velocity of rapid regressors (4.58 ± 4.43 cc per fraction) was significantly (*p* < 0.001) higher than that of slow regressors (0.78 ± 0.40 cc per fraction). The mean DFS of rapid regressors was 36.8 months, longer than the 31.9 months of slow regressors (*p* = 0.400) without statistical significance. Rapid regressors had superior DFS rate compared to slow regressors, although statistically insignificant (Figure 2). The 2-year DFS was 88.9% for rapid regressors and 72.7% for slow regressors, respectively (*p* = 0.400).

The tumor regression rate was compared between rapid regressors (72.58 ± 17.99%) and slow regressors (64.98 ± 14.61%). Although rapid regressors did have higher tumor regression rate, the difference was statistically insignificant (*p* = 0.272). When the tumor regression rate was analyzed for correlation with DFS, it failed to discriminate DFS. Median tumor regression rate was 73.6% in all patients. There was no difference in DFS of patients with tumor regression rate > 73.6% (77.8% at 2 years) and patients with tumor regression rate ≤ 73.6% (80% at 2 years).

## 4. Discussion

This study is the first to demonstrate a steady, linear pattern of tumor volume regression by daily measurement of rectal tumor with MRI during preoperative CCRT in locally advanced rectal cancer. The primary endpoint of this study showed linear tumor regression by 2.4 cc per fraction with high statistical significance. When the patients were divided into rapid regressors and slow regressors according to the tumor regression velocity (cc) per fraction, rapid regressors had significantly higher tumor regression velocity (4.58 cc per fraction) compared to slow regressors (0.78 cc per fraction). The secondary endpoint of this study demonstrated that tumor biology and real-time treatment sensitivity represented by tumor regression velocity was significantly different between rapid and slow regressors. Rapid regressors did have a longer DFS compared to slow regressors, but the difference was statistically insignificant.

Although the DFS of rapid regressors was superior to that of slow regressors without statistical significance, this study has demonstrated the prognostic potential of tumor regression velocity. The simultaneous assessment of tumor response to radiation and chemotherapy on a daily basis provides an advanced insight on the tumor biology, which enables patient-driven treatment decisions at a very early phase: prior to surgery and even before preoperative MRI. According to our data, tumor regression rate failed to discriminate tumor outcome, while daily tumor regression velocity demonstrated difference in DFS, although without statistical significance. Despite the fact that further validation of the efficacy of tumor regression velocity as a predictor of DFS is necessary, our results suggest the potential of tumor regression velocity in guidance to patient-driven cancer therapy, such as consideration of treatment intensification for slow regressors and vice versa.

However, several issues raised by the result of the primary endpoint of this study need to be considered first. The apparent pattern of tumor volume regression was linear, which seems disparate from previous experimental observations that cell killing occurs in an exponential function of dose. With regards to the most widely accepted linear quadratic model, which may be simplified as a model showing two components of cell killing by radiation (linear referring to the component proportional to dose and quadratic referring to the other component proportional to the square of dose), these data may be interpreted as demonstrating the linear phase of tumor regression [27]. However, this study is hypothesis-generating and whether the observed linearity of tumor volume regression will be followed by exponential tumor regression as in the linear quadratic model needs further validation.

In accounting for possible mechanisms under the linear tumor volume regression during CCRT, a comprehensive review of tumor biology may shed some light. A reminder on the constituents of a tumor can be a starting point. Malignant tumor is more than a pure assortment of tumor cells. Instead, it is a heterogeneous composition of vascular, stromal, and immune cells within the extracellular matrix containing various types of secreted proteins in addition to tumor cells [28]. In vivo tumor volume is intrinsically different from in vitro tumor cell lines, which exhibit exponential survival curve. The endpoint for cells cultured in vitro is the loss of clonogenic potential rather than actual cell death. Although a tumor cell may have lost its clonogenic potential by irradiation, tumor volume may stay the same if the cell remains in the tumor bulk. Physical volume reduction in a tumor is the result of actual cell loss rather than its loss of clonogenic potential.

There are several possible ways of cell loss besides direct cell killing by CCRT: metastasis, inadequate nutrition due to poor vasculature and tumor outgrowth from its vascular dimension, apoptosis, and immune attack [29,30,31]. In case of rectal cancer, cell loss by exfoliation of necrotic debris is also possible. Normal cells within the tumor may also be lost collaterally due to similar reasons. In addition to loss of tumor and normal cells, the process of tumor volume reduction includes the clearance of dead materials through lymphatic drainage with the help of scavenger immune cells. The rectum has rich lymphatics, which are likely to show more rapid tumor volume reduction [32]. It is a well-vasculated organ, resulting in better oxygenation and sensitization of tumor cells to treatment [33]. In addition to the organ-specific characteristics of primary tumor site, the intrinsic radiosensitivity of individual tumors may influence tumor volume kinetics during treatment. The results of this study are specifically from rectal adenocarcinoma with mean daily tumor volume regression velocity of 2.4 cc. However, the radiosensitivity of each tumor varies [34], observed as different tumor volume regression velocities between patients.

There are treatment factors to consider as well. The chemotherapeutic regimen for CCRT in this study was daily oral administration of capecitabine. The radiosensitizing effect of daily capecitabine may have led to the pronounced linear pattern of tumor volume regression compared to IV leucovorin and fluorouracil given at the first and fifth weeks of CCRT. In terms of radiation, the conventional fraction of 2 Gy per day was delivered in a long course of 5 weeks. The variation in the pattern of tumor volume regression according to different chemotherapeutic and fractionation regimens remain to be seen.

In addition to the possible mechanisms for linear tumor regression (primary endpoint) described above, the feasibility of tumor regression velocity as a predictor of DFS (secondary endpoint) also needs to be reviewed. Because this study is a retrospective analysis of a small cohort, data were insufficient to show statistically significant difference in DFS of rapid and slow regressors. Capecitabine was administered during CCRT, while other options were allowed for adjuvant chemotherapy. However, the prognostic potential of daily tumor regression velocity has been primitively suggested by discrimination of DFS, although statistically insignificant. It may have been better demonstrated in this study due to more precise observation enabled by the superior soft tissue contrast of MRI compared to conventional registration CT images [22]. Daily tumor regression velocity obtained from daily MRIs throughout the full course of CCRT may provide more accurate information on individual tumor biology and prognosis compared to preoperative MRI or weekly MRI during CCRT. Daily tumor volumetry during CCRT yields 25 measurements of tumor volume for each patient. Compared to previous studies on preoperative MRI (two measurements of tumor volume per patient from MRIs at pre-/post-CCRT) or weekly MRI during CCRT (up to seven measurements of tumor volume per patient), this study obtains significantly greater number of observations per patient. By virtue of more observations, daily tumor volumetry may surpass weekly MRI in reflecting biology, treatment response, and the prognosis of a tumor [20]. A previous study, which performed daily MRI (171 scans) on eight patients showed different tumor volume dynamics between patients with pathological partial response (pPR) and pathological complete response (pCR) [35]. Instead of using a pathologic surrogate endpoint, we demonstrated that tumor regression velocity may potentially segregate DFS by analyzing a total of 500 sessions of MRI. Contouring was performed by single physician with reference to contouring atlas in order to minimize inter- and intra-observer variability [36,37]. This study is an observation of on-site rectal adenocarcinoma while it is still inside a patient enabled by a newer technology. Despite dramatic advances in experimental techniques mimicking real life discrepancies, such as organoids and numerous other model tumor systems, between bench and clinic persist [38]. Even the most exquisite patient-derived in vivo models still have their own limitations, only because they are not the individual patients themselves [39].

This is the first report of linear regression of on-site rectal tumor conserved in its entirety during CCRT using daily MRI. We anticipated to take the first steps towards disclosing the clinical value of tumor regression velocity in this pilot study. However, the statistically insignificant difference in DFS by tumor regression velocity needs further validation if the difference becomes statistically significant with further follow-up of additional patients. The macroscopically linear pattern of tumor volume regression needs further exploration for its relation to microscopic tumor dynamics. Whether daily tumor volume regression velocity may serve as a feasible tool for the decision of treatment intensification or de-intensification remains to be seen because the practical value concerning correlations to clinical parameters requires further validation. Considering the labor and time intensive character, as well as the cost of tumor volumetry using MRI, its pragmatic value remains to be seen.

## 5. Conclusions

Collectively, this study is the first observation of linear tumor regression in daily MRI during preoperative CCRT of locally advanced rectal cancer. Daily tumor regression velocity discriminated DFS, although without statistical significance. This study with a phenomenal approach is hypothesis-generating. Nevertheless, the potential of CCRT from therapeutics to a newer level, “theranostics”, has been inceptively suggested. Further validation studies for the value of daily tumor volumetry on treatment decisions are warranted.

## Figures and Tables

**Figure 1 cancers-14-03749-f001:**
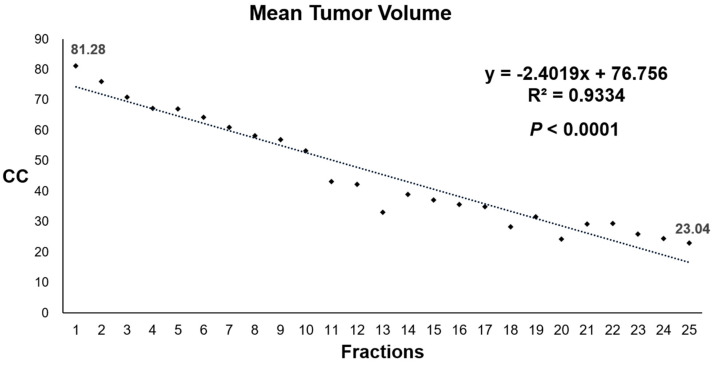
Mean values of absolute tumor volume (cc) at each fraction of 20 patients who underwent chemoradiotherapy.

**Figure 2 cancers-14-03749-f002:**
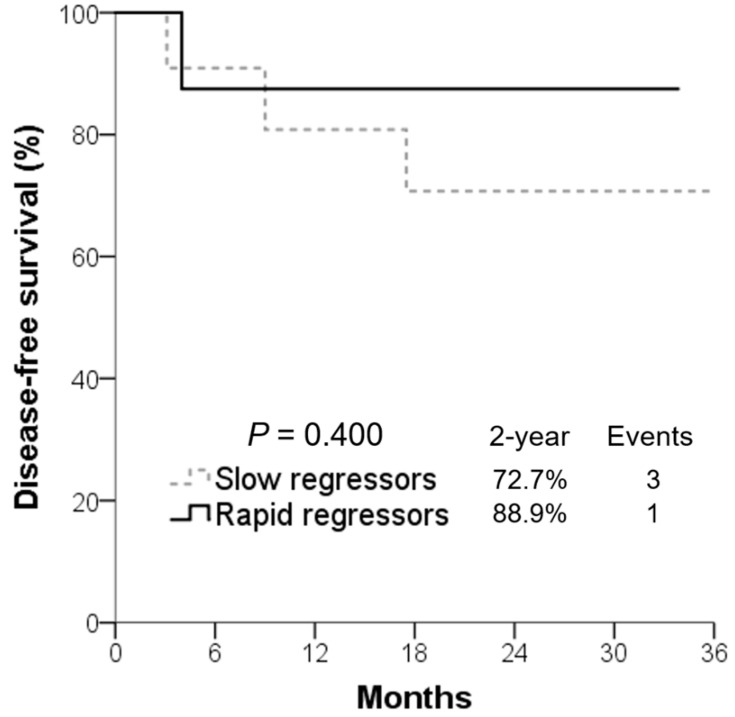
Rapid regressors had numerically superior disease-free survival rate compared to slow regressors.

**Table 1 cancers-14-03749-t001:** Tumor, patient, and treatment characteristics of 20 patients who underwent chemoradiotherapy.

Total N = 20 (%)		Groups According to Tumor Regression Velocity		*p*
		Rapid Regressors N = 9 (%)	Slow Regressors N = 11 (%)	
**Tumor regression velocity (cc) per fraction**				<0.001
R^2^ (*p*)	2.40 0.93 (<0.001)	4.58 0.90 (<0.001)	0.78 0.97 (<0.001)	
Mean	2.49	4.58	0.78	
SD	3.48	4.43	0.40	
Median	1.52	3.18	0.85	
Range	0.23–14.13	1.52–14.13	0.23–1.41	
**Age** (median 64, range: 47–87)				1.000
<64	11 (55)	4 (44.4)	5 (45.5)	
≥64	9 (45)	5 (55.6)	6 (54.5)	
**Sex**				0.479
Male	18 (90)	9 (100)	9 (81.8)	
Female	2 (10)	0 (0)	2 (18.2)	
**Clinical T stage**				0.642
T3	13 (65)	5 (55.6)	8 (72.7)	
T4	7 (35)	4 (44.4)	2 (27.3)	
**Clinical N stage**				0.370
N0	1 (5)	0 (0)	1 (9.1)	
N1	9 (45)	3 (33.3)	6 (54.5)	
N2	10 (50)	6 (66.7)	4 (36.4)	
**Tumor size** (median 5, range: 3–9)				0.092
≤5 cm	10 (50)	2 (22.2)	7 (63.6)	
>5 cm	10 (50)	7 (77.8)	4 (36.4)	
**Anal verge** (median 6.8, range: 1–12)				0.591
≤5 cm	5 (25)	1 (11.1)	3 (27.3)	
>5 cm	15 (75)	8 (88.9)	8 (72.7)	
**CEA** (median 8, range: 1–92.3 ng/mL)				0.642
<3.5 ng/mL	6 (30)	2 (22.2)	4 (36.4)	
≥3.5 ng/mL	14 (70)	7 (77.8)	7 (63.6)	
**Differentiation**				1.000
Well	3 (15)	1 (11.1)	2 (18.2)	
Moderate	13 (65)	5 (55.6)	8 (72.7)	
Unavailable	4 (20)	3 (33.3)	1 (9.1)	
**Response rate** (RECIST v1.1)				0.770
Complete response	1 (5)	0 (0)	1 (9.1)	
Partial response	16 (80)	7 (77.8)	9 (81.8)	
Stable disease	3 (15)	2 (22.2)	1 (9.1)	
**Resection rate**				1.000
Surgery	14 (70)	6 (66.7)	8 (72.7)	
Observation	6 (30)	3 (33.3)	3 (27.3)	
**Adjuvant chemotherapy**				0.854
FOLFOX	6 (30)	2 (22.2)	4 (36.4)	
Capecitabine	7 (35)	3 (33.3)	4 (36.4)	
None	7 (35)	4 (44.4)	3 (27.3)	
**Resected Patients** **N = 14 (%)**		**Rapid Regressors** **N = 6 (%)**	**Slow Regressors** **N = 8 (%)**	***p***
**yp T stage (N = 14)**				0.385
T1	2 (14.3)	1 (16.7)	1 (12.5)	
T2	1 (7.1)	1 (16.7)	0 (0)	
T3	10 (71.4)	3 (50)	7 (87.5)	
T4	1 (7.1)	1 (16.7)	0 (0)	
**yp N stage (N = 14)**				0.083
N0	7 (50)	6 (100)	3 (37.5)	
N1	6 (42.9)	0 (0)	2 (25)	
N2	1 (7.1)	0 (0)	3 (37.5)	
**Lymphatic invasion (N = 14)**				1.000
Yes	1 (7.1)	0 (0)	1 (12.5)	
No	13 (92.9)	6 (100)	7 (87.5)	
**Vascular invasion (N = 14)**				0.209
Yes	3 (21.4)	0 (0)	3 (37.5)	
No	11 (78.6)	6 (100)	5 (67.5)	
**Perineural invasion (N = 14)**				0.627
Yes	6 (42.9)	2 (33.3)	4 (50)	
No	8 (57.1)	4 (66.7)	4 (50)	
**Margin (N = 14)**				1.000
Positive	2 (14.3)	1 (16.7)	1 (12.5)	
Negative	12 (85.7)	5 (83.3)	7 (87.5)	

Abbreviations: CEA = carcinoembryonic antigen, RECIST = response evaluation criteria in solid tumors, FOLFOX = leucovorin + fluorouracil + oxaliplatin.

**Table 2 cancers-14-03749-t002:** Tumor regression velocity (cc) per fraction, tumor regression rate (%), and disease-free survival (DFS) of rapid regressors and slow regressors.

Group According to Tumor Regression Velocity	Patient (N = 20)	Tumor Regression Velocity (cc) per Fraction	Tumor Regression Rate (%)	DFS (Months)
**Rapid regressors** **(N = 9)**	1	14.13	36.98	40.5
	2	9.87	74.67	7.5
	3	3.96	73.60	36.1
	4	3.35	31.70	31.6
	5	3.18	74.39	32.6
	6	1.98	80.88	33.7
	7	1.65	71.38	36.0
	8	1.59	94.06	38.2
	9	1.52	79.99	35.6
	Mean SD (SE)	4.58 4.43	72.58 17.99	36.8 3.5
**Slow regressors** **(N = 11)**	10	1.41	59.79	9.5
	11	1.10	49.11	21.2
	12	1.07	79.21	34.0
	13	1.05	47.76	33.4
	14	1.02	70.06	38.0
	15	0.85	64.13	39.3
	16	0.82	80.94	34.5
	17	0.46	67.12	37.7
	18	0.36	74.52	6.3
	19	0.26	82.90	32.4
	20	0.23	39.29	36.5
	Mean SD (SE)	0.78 0.40	64.98 14.61	31.9 3.8
*p*		<0.001	0.272	0.400

## Data Availability

The data presented in this study are available on request from the corresponding author.

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
