# Peer review of "Linear Tumor Regression of Rectal Cancer in Daily MRI during Preoperative Chemoradiotherapy: An Insight of Tumor Regression Velocity for Personalized Cancer Therapy"

_cancers, 2022, doi:10.3390/cancers14153749_

Round 1

Reviewer 1 Report

The study obviously is original, thus contributing to theory of determining radiation response characteristics and for tumor biology. Although, the authors shed a lot of hopes on the practical aspect in treating rectal cancer patients with effective radiochemotherapy eventually leading to skip surgery, which is  expressed in the introduction. This aim is not reached at all and discussed later in the "Discussion-Section".

There is no correlation to DFS and Resection/No Resection. The decisions for surgery "yes" or "no" obviously were made by the patients. The "hopes" for clinical relevance (in the future) are based on "numerical" differences. The term "numerical" is misleading. In the literature this corresponds to "strong or mild tendencies". 

Some comments:

1. It is not clear, whether the rectal cancers were located in the lower two thirds of the rectum.

2. Avoid the term "numerically".

3. Are the patients selected or did you do your study in consecutive patients treated and observed by you?.

4. Did you correlate the dynamics of CEA-decline to your parameters in "primary aim" and "secondary aim".

5. The value of the results concerning "tumor biology" are important and well discussed.

6. In the sections "M+M" and "Discussion" you make the statement that all patients received "capecitabine" for CT. However, for adjuvant therapy, other options are chosen, as well. This might be mentioned.

7. The practical value concernig correlations to clinical parameters remains to be shown, unfortunately. Please make this very clear and avoid the term "numerical".

8. How would you design a more promising study? 

8. Could you give an idea of the costs?

Author Response

The study obviously is original, thus contributing to theory of determining radiation response characteristics and for tumor biology. Although, the authors shed a lot of hopes on the practical aspect in treating rectal cancer patients with effective radiochemotherapy eventually leading to skip surgery, which is expressed in the introduction. This aim is not reached at all and discussed later in the "Discussion-Section".

Authors’ response:

Thank you for your perceptive comment. What we intended was to suggest the POTENTIAL value of tumor regression velocity in decision of treatment intensification versus de-intensification at an early phase of treatment course. However, as you have pointed out, the expressions used in Introduction and Discussion do seem to be misleading as proposing tumor regression velocity as a DEFINITIVE prognosticator for omission of surgery. Thus, we have rephrased Introduction and Discussion in order to proffer tumor regression velocity as a biomarker with more weight on tumor biology and treatment response as follows:

Introduction:

“However, pathologic response can be assessed only after surgery and then it would be too late to provide information on whether to omit surgery or not.”

à Rephrased: “However, pathologic response can be assessed only after surgery.”

“…real-time treatment sensitivity and tumor biology TO guide patient-driven treatment decision.”

à Rephrased: “… real-time treatment sensitivity and tumor biology WHICH MAY POTENTIALLY guide patient-driven treatment decision.”

Discussion:

“Whether daily tumor volume regression velocity may serve as a feasible tool for decision of treatment intensification or de-intensification remains to be shown because the practical value concerning correlations to clinical parameters requires further validation.”

There is no correlation to DFS and Resection/No Resection. The decisions for surgery "yes" or "no" obviously were made by the patients. The "hopes" for clinical relevance (in the future) are based on "numerical" differences. The term "numerical" is misleading. In the literature this corresponds to "strong or mild tendencies". 

Authors’ response:

We deeply appreciate your thorough and comprehensive review which markedly enhances the quality of the manuscript. We recognize the validity of your points and addressed them appropriately as follows:

1. Added the following sentence in Results:

There was no difference in DFS (P = 0.730) between the resected patients (2-year 78.6%) and those who did not undergo resection (2-year 83.3%).

2. The term “numerically” has been replaced with “statistically insignificant” or “without statistical significance” throughout the manuscript.

Some comments:

  1. It is not clear, whether the rectal cancers were located in the lower two thirds of the rectum.

Authors’ response:

Thank you for pointing out an important aspect of rectal cancer. It is true that neoadjuvant CCRT is more prioritized for rectal cancers located in the lower two thirds of the rectum. However, we measured the lower border of a tumor from anal verge (AV). The tumors we treated were located median 6.8 cm above AV, which is within the lower two thirds of the rectum. The range of tumor locations from AV was 1 – 12 cm, so the tumors we treated were low-lying tumors. However, we did not exclude a tumor located on the upper one third of the rectum on purpose just because of its location if its locally aggressive features may benefit from CCRT.

  1. Avoid the term "numerically".

Authors’ response:

Thank you for eliminating the confusions which the manuscript may have generated. The term “numerically” has been replaced with “statistically insignificant” or “without statistical significance” throughout the manuscript.

  1. Are the patients selected or did you do your study in consecutive patients treated and observed by you?

Authors’ response:

As described in the Patients section of Material and Methods, “Patients diagnosed with locally advanced rectal cancer who underwent preoperative concurrent chemoradiotherapy (CCRT) at a tertiary medical institution from November 2018 to June 2019 were CONSECUTIVELY included.”

  1. Did you correlate the dynamics of CEA-decline to your parameters in "primary aim" and "secondary aim".

Authors’ response:

Thank you for your insightful comment. We have not yet thought of correlating the dynamics of CEA with the tumor volume dynamics. We only obtained the pretreatment CEA levels. Unfortunately, the post-CCRT levels of CEA are currently unavailable due to practical reasons (the author who provided this data has currently taken a leave of absence). We do recognize the scientific significance of your comment and are strongly willing to include CEA dynamics into our future studies. We once again deeply appreciate your constructive remark.

  1. The value of the results concerning "tumor biology" are important and well discussed.

Authors’ response:

Thank you for recognizing our eagerness to elucidate the biologic rationale under the newly observed phenomenon of linear tumor regression.

  1. In the sections "M+M" and "Discussion" you make the statement that all patients received "capecitabine" for CT. However, for adjuvant therapy, other options are chosen, as well. This might be mentioned.

Authors’ response:

We appreciate your thorough review. We have mentioned the above details in Method and Discussion as follows.

Method:

“Adjuvant systemic therapy after surgery was administrated on the physicians’ discretion.”

Discussion:

“Capecitabine was administered during CCRT while other options were allowed for adjuvant chemotherapy.”

  1. The practical value concerning correlations to clinical parameters remains to be shown, unfortunately. Please make this very clear and avoid the term "numerical".

Authors’ response:

Thank you for clarifying the thesis of this manuscript. The term “numerical” in the manuscript were replaced with “statistically insignificant” or “without statistical significance”. We have included the above information in the Discussion as follows:

“Whether daily tumor volume regression velocity may serve as a feasible tool for decision of treatment intensification or de-intensification remains to be shown because the practical value concerning correlations to clinical parameters requires further validation.”

  1. How would you design a more promising study? 

Authors’ response:

We envisioned the following studies:

1. Clinical study: First, we would further maturate our data and reanalyze by including more patients with longer follow-up. As you have previously suggested, we would like to identify any correlation between tumor regression velocity and CEA level dynamics. With a larger cohort, more thorough assessment is possible by implementing various analytic methods such as Cox regression analysis to adjust for other biomarkers, such as CEA levels.

2. Animal study: We would like to set up an animal model mimicking human CCRT with daily MRI, in compliance with the ethics of animal research. We would like to observe the daily cell composition change of animal rectal tumor from day 1 to day 25 of CCRT and analyze its correlation with macroscopic tumor volume dynamics.

However, we stated the future directions as follows due to space limitations:

“We anticipated to take the first steps towards disclosing the clinical value of tumor regression velocity in this pilot study. However, the statistically insignificant difference in DFS by tumor regression velocity needs further validation if the difference becomes statistically significant with further follow-up of additional patients. The macroscopically linear pattern of tumor volume regression needs further exploration for its relation to microscopic tumor dynamics.”

  1. Could you give an idea of the costs?

Authors’ response:

Thank you for your pragmatic comment.

1. Machine-wise, MR-LINAC is approximately 2 to 3 times more expensive than CT-LINAC.

2. Utilization rate-wise, because of the time consumed in obtaining MR images, the number of patients per day treatable with MR-LINAC is approximately 60% compared to that of CT-LINAC.

3. Human resource-wise, the radiation oncologist dedicated for delineation of 25 sets of MRI per patient may otherwise handle 25 new patients.

Therefore, tumor volumetry using daily MRI is far from a cost-effective approach and rather a result of cutting-edge technology combined with specialized personnel. However, the above obstacles could be overcome with rapidly developing technologies. With further technological advances, the cost of MR-LINAC will be lowered; the utilization rate of MR-LINAC will improve; and artificial intelligence-assisted auto-segmentation will delineate daily MRIs. We anticipate that the results of this pioneering study in the present era will contribute to more personalized cancer therapy in the coming generation. Due to space limitations, we have included the above aspects in the Discussion as follows:

“Considering the labor and time intensive character as well as the cost of tumor volumetry using MRI, its pragmatic value remains to be seen.”

Reviewer 2 Report

Dear Authors

I’ve read your article with great interest and I’m very impressed by the scope of your work. My criticism is following. 

1.     Surgery remains the cornerstone in achievement of Local control of rectal cancer and CRT could not compensate suboptimal surgical technique. Though you mention R status of surgery I think the data about operation should be provided. At least  the rate of APR, TME or partial TME

If we look at comparison of ypTN it is obvious that slow regressors had more aggressive tumors, i.e. more LN mts, positive vascular, lymphatic and perineural invasion. It makes conclusion of better DFS in rapid regressors controversial. Presumably it can be related to the effect of CRT, though rapid regressors had less advanced carcinomas before treatment. This issue should be addressed

Author Response

I’ve read your article with great interest and I’m very impressed by the scope of your work. My criticism is following. 

Authors’ response:

Thank you very much for the encouraging comment. Your comments are well-received and responses are given point-by-point as below.

      Surgery remains the cornerstone in achievement of Local control of rectal cancer and CRT could not compensate suboptimal surgical technique.

Authors’ response:

1. We appreciate your constructive comment. What we intended was to suggest the POTENTIAL value of tumor regression velocity in decision of treatment intensification versus de-intensification at an early phase of treatment course. However, as you have pointed out, the expressions used in Introduction and Discussion do seem to be misleading as proposing tumor regression velocity as a DEFINITIVE prognosticator for omission of surgery. Thus, we have rephrased Introduction and Discussion in order to proffer tumor regression velocity as a biomarker with more weight on tumor biology and treatment response as follows:

Introduction:

“However, pathologic response can be assessed only after surgery and then it would be too late to provide information on whether to omit surgery or not.”

à Rephrased: “However, pathologic response can be assessed only after surgery.”

“…real-time treatment sensitivity and tumor biology TO guide patient-driven treatment decision.”

à Rephrased: “… real-time treatment sensitivity and tumor biology WHICH MAY POTENTIALLY guide patient-driven treatment decision.”

Discussion:

“Whether daily tumor volume regression velocity may serve as a feasible tool for decision of treatment intensification or de-intensification remains to be shown because the practical value concerning correlations to clinical parameters requires further validation.”

Though you mention R status of surgery I think the data about operation should be provided. At least the rate of APR, TME or partial TME.

Authors’ response:

Thank you for pointing out a very important detail. We have added the following sentence in the Results section as follows:

“Among 14 patients who underwent surgery, 9 had laparoscopic low anterior resection with ileostomy, 4 received intersphincteric resection, and 1 had conversion Hartmann’s operation.”

If we look at comparison of ypTN it is obvious that slow regressors had more aggressive tumors, i.e. more LN mts, positive vascular, lymphatic and perineural invasion. It makes conclusion of better DFS in rapid regressors controversial. Presumably it can be related to the effect of CRT, though rapid regressors had less advanced carcinomas before treatment. This issue should be addressed.

Authors’ response:

Thank you for commenting a very crucial issue. Just as you have pointed out, slow regressors had more aggressive features than rapid regressors after CCRT in surgical pathology. However, rapid regressors had clinically more aggressive disease at initial diagnosis compared to slow regressors. Rapid regressors had more T4 (rapid 44.4% vs. slow 27.3%) and more N2 (rapid 66.7% vs. slow 36.4%) disease. Therefore, we considered the correlation of tumor regression velocity with DFS was reasonable, although needs further validation. Because the conversion of disease severity of rapid regressors from more aggressive to less aggressive between CCRT is not so apparent, we have added the following sentences in the Results section:

“Before CCRT, rapid regressors had more aggressive disease compared to slow regressors. Although statistically insignificant, rapid regressors had more T4 (rapid 44.4% vs. slow 27.3%) and more N2 (rapid 66.7% vs. slow 36.4%) disease. After CCRT, slow regressors had more aggressive features compared to rapid regressors including more T3 (rapid 50% vs. slow 87.5%), more N1/2 (rapid 0% vs. slow 63.5%), more vascular and perineural invasions, although without statistical significance.”